# Negative Parenting Styles and Psychological Crisis in Adolescents: Testing a Moderated Mediating Model of School Connectedness and Self-Esteem

**DOI:** 10.3390/bs13110929

**Published:** 2023-11-15

**Authors:** Jinguo Zhao, Haiyan Zhao, Aibao Zhou

**Affiliations:** School of Psychology, Northwest Normal University, Lanzhou 730070, China; 2021104103@nwnu.edu.cn (J.Z.); 2021104113@nwnu.edu.cn (H.Z.)

**Keywords:** rejecting parenting style, controlling parenting style, self-esteem, school connectedness, adolescents

## Abstract

Little is known about how rejecting and controlling parenting styles may influence adolescent psychological crisis and what conditions may buffer the detrimental effects of psychological crisis. By integrating multiple theories, this study investigated self-esteem as an underlying mediator and school connectedness as a potential moderator to explain the link between negative parenting and the two psychological crises among Chinese adolescents. In this study, the questionnaire method is adopted to examine the combined mechanism of rejecting parenting style and controlling parenting style on the psychological crisis of adolescents. In total, 1863 adolescents were involved in this study, ranging from 13 to 17 years old. The results showed that both the rejecting parenting style and the controlling parenting style can significantly and positively predict the level of psychological crisis of adolescents, but the predictive power of the rejecting parenting style was stronger. Self-esteem partially mediates the relationship between rejecting parenting style, controlling parenting style, and psychological crisis. School connectedness moderates both the effects of rejecting parenting styles on self-esteem and the effects of self-esteem on the psychological crisis. This study identifies the internal mechanisms by which negative parenting styles affect adolescents’ psychological crisis, and reveals the mediating and moderating roles of self-esteem and school connectedness, providing additional explanatory paths for the mechanisms of adolescents’ psychological crisis.

## 1. Introduction

Adolescents are at high risk of a psychological crisis [1] A psychological crisis is a series of psychologically imbalanced reactions that occur when an individual’s usual resources are insufficient to cope with a setback [2]. Depending on the behavioral outcome triggered, psychological crises have been classified as non-deadly and deadly [3]. Non-deadly crises include emotional disorders and problem behaviors such as anxiety, depression, and Internet addiction, while deadly crises include self-harm and suicidal behaviors. Depression and non-suicidal self-harming behaviors are considered typical manifestations of psychological crisis in adolescents due to their high prevalence and risk [4,5]. In China, annual depression and non-suicidal self-harming prevalence rates of over 21% and 29% among adolescents have been reported [6,7], which is much higher than that worldwide [8,9]. Numerous studies have shown that both depression and non-suicidal self-harming behavior are significant predictors of suicidal behavior in adolescents [10,11,12]. Moreover, there is a high correlation and co-morbidity between depression and non-suicidal self-harming behaviors. The more severe the depressive symptoms in adolescents, the more self-harming behaviors they have [13]. Also, adolescents’ past self-harming behaviors significantly predict later depression [14]. Therefore, it is necessary to study both depression and non-suicidal self-harming behaviors as important indicators of psychological crisis to enrich related theories.

The emergence of psychological crisis is, on the one hand, due to the imbalance in the development of psychological, cognitive, physiological, and other individual factors during adolescence; on the other hand, it is the result of the interaction of complex environmental factors such as school, peers, family, and community [15]. According to the ecosystem theory [16], family and school are important environmental microsystems for adolescents [17]. Among family environmental factors, parenting styles have significant impacts on adolescents’ cognition, emotions, and behaviors [18]. Research has found that the rejecting parenting style and the controlling parenting style, the two most common negative parenting styles [19], significantly predicted depression [20,21] and self-harming behavior [22,23] in adolescents. The rejecting parenting style refers to the parenting style in which parents tend to be rejecting and punitive with their children [24], while the controlling parenting style, also known as overprotection, refers to excessive parental control over children’s daily activities and behaviors, encouraging children to rely on their parents [25]. The self-determination theory [26] emphasizes that individuals are prone to poor coping and negative emotions when the external environment fails to satisfy their three basic psychological needs: a sense of competence, a sense of autonomy, and a sense of relatedness. Some studies have found that the rejecting parenting style will cut off the emotional connection between parents and children, so that children’s emotional needs cannot be met, and the children are prone to depression and self-harm [27,28]. When parents have too much control over their children, adolescents’ autonomy needs are not met [29], which in turn leads to internalization problems, showing higher levels of depression and self-harming [30,31,32]. Both of these types of negative parenting styles can lead to psychological and social maladjustment, further causing psychological crisis. Furthermore, parents may not be limited to one particular form of parenting [33], and many parents who use negative parenting not only show a high degree of control but also demand unconditional obedience from their children [34]. However, most of the existing research understands the two parenting styles, rejecting and controlling, as two isolated dimensions and examines their relationship with other psychological and behavioral variables separately, which does not provide a comprehensive and realistic picture of the complex parenting situation. Therefore, the mechanism of the combined effect of rejecting parenting and controlling parenting on adolescent psychological crisis needs to be further elucidated.

### 1.1. Self-Esteem as a Mediator

Family environmental factors often work through the internal factors of an individual. Self-esteem is the integration of an individual’s perceptions of value, competence, and self-acceptance, reflecting the individual’s self-perception [35]. According to the susceptibility model [36], low self-esteem is a risk factor leading to depression in individuals. When individuals are in a negative cognitive state (e.g., they believe they are worthless), they are more likely to adopt a pessimistic view of the future, which can lead to depression [37]. Low levels of self-esteem predict high levels of depression and non-suicidal self-harming [38,39,40]. Conversely, when individuals have positive perceptions of themselves, they can alleviate depression and protect their mental health through greater self-affirmation [41].

Adolescence is the peak period of an individual’s personality development and a critical period for the development of self-esteem. Tafarodi and Swann Jr (1995) suggest that the sense of value and the sense of competence constitute the most basic structures of self-esteem [42]. A sense of competence, from the individual’s experience of success, failure, influence, and mastery, is a subjective evaluation of the self’s ability to cope with challenges in life, which influences the individual’s sense of control, efficacy, and other aspects of the individual’s behavioral autonomy, while a sense of value is a subjective evaluation of the degree to which the self conforms to the standards of social value, which is derived from the individual’s experience of acceptance and rejection in interpersonal interactions, morality, and guilt, and it is also a reflection of an individual’s sociality. From the interpersonal perspective, Mark Leary proposed a new interpretation of self-esteem, The Sociometer Theory of Self-Esteem, which suggests that an individual’s self-esteem changes with different degrees of acceptance from others in the social environment [43]. Numerous studies have shown that parental acceptance is strongly associated with high self-esteem in children, whereas negative parenting styles, such as parental psychological control and rejection, can hinder the development of self-esteem to varying degrees, creating low self-esteem in children [44,45,46]. Specifically, rejecting parenting styles can make children feel that they are not accepted and recognized by their parents, resulting in a low sense of self-worth [47], which in turn leads to lower self-esteem. Similarly, high levels of parental control over their children can make adolescents lack autonomy and independence [48], which in turn triggers a sense of low competence [49] and lowers self-esteem. It can be hypothesized that self-esteem may mediate the effect of the rejecting parenting style and the controlling parenting style on psychological crisis.

### 1.2. School Connectedness as a Moderator

Although research has shown that negative parenting styles influence the occurrence of psychological crisis in adolescents, not all adolescents who have rejecting and controlling parenting experiences have a psychological crisis; they can still achieve resilient and positive outcomes, which may be moderated by protective factors. The buffering theory of social support [50] states that when individuals experience risk factors (e.g., rejection or controlling parenting), social support enables individuals to experience more positive emotions and reduces the adverse effects of the risk factors. For adolescents, the influence of the school environment on their physical and mental health gradually increases and the influence of the home environment weakens [51]. In the daily academic life of adolescents, in addition to their parents, others with high interaction including teachers and peers have a significant impact on adolescents’ psychology [52]. School connectedness has been defined as having access to caring and supportive interpersonal relationships at school [53]. Students with higher levels of school connectedness typically have a sense of belonging and liking for school and have positive relationships with their peers and teachers [54,55,56]. A study by Chionh and Fraser (2009) found that school teacher support predicted students’ self-esteem [57], while the study by Gentina (2018) demonstrated that peer support for middle school students significantly enhanced feelings of self-worth and improved self-esteem [58]. So the higher the individual school connection, the higher the individual’s level of self-esteem.

Individuals with high self-esteem possess a better sense of control over their environment and can effectively utilize the social support in their environment to buffer themselves when coping with stressful events, reducing the negative impact of stressful events on them [59]. Although negative parenting styles can reduce individuals’ self-esteem levels, high school connectedness gives individuals more emotional support, thus buffering the effects of both negative parenting styles on self-esteem. On the other hand, when presented with stimulus information, high self-esteem individuals can better filter negative information and are prone to perceive more social support [60].

### 1.3. The Present Study and Hypotheses

To sum up, with a large sample of adolescents, this study examines the mechanism of the rejecting parenting style and the controlling parenting style on psychological crisis. The hypotheses of this study are as follows:

**Hypothesis** **1.**
*Rejecting parenting styles and controlling parenting styles not only directly predict adolescent psychological crisis, but also influence adolescent psychological crisis through self-esteem.*


**Hypothesis** **2.**
*School connectedness not only moderates the effects of rejecting and controlling parenting on adolescent self-esteem but also acts on the effects of self-esteem on the psychological crisis.*


## 2. Methods

### 2.1. Participants and Procedure

Convenience sampling was used in this study. With the assistance of local education authorities in a mid-sized city located in northwestern China, four public middle and high schools were randomly invited to participate in our research between March and April 2023. With the informed consent of school leaders, teenagers, and their parents, a total of 2161 students agreed to participate in the survey and completed it during one of their normal classes with the administration of a trained research assistant. All questionnaires were filled out anonymously, and participants were required to truthfully and independently answer each question. Finally, the number of valid questionnaires was 1863 (86.2%). Among the participants, 974 (52.3%) were male and 889 (47.7%) were female. The age of the participants ranged from 13 years to 17 years, in early and mid-adolescence, with a mean age of 13.50 ± 1.38 years. Based on parent-reported information, almost all the participants were from middle-income families with parents who had earned at least a middle school degree. Nearly all respondents were living with their parents. The Institutional Review Board (IRB) of the authors’ university reviewed the research protocol to guarantee that the research ethics were met.

### 2.2. Measures

#### 2.2.1. Parenting Style Scale

The Short Egna Minnen av Barndoms Uppfostra Questionnaire [61] was used to assess the negative parenting styles. It includes three subscales, namely rejection, emotional warmth, and overprotection. According to the research needs, two subscales of “rejection” and “overprotection” were selected for measurement in this study. Responses were coded based on a 4-point Likert scale, with response options ranging from 0 (never) to 4 (always). The scale has shown adequate reliability and validity with Chinese adolescents [62]. The Cronbach’s alpha coefficients of two sub-scales of parenting styles were 0.89 (rejection) and 0.87 (over-protection), respectively.

#### 2.2.2. Self-Esteem Scale

The Rosenberg Self-Esteem Scale [63] was used to measure the self-esteem level of the participants. The original scale consists of 10 questions and is divided into two dimensions: self-affirmation and self-negation, and the self-affirmation dimension with higher factor loadings was chosen for this study. The scale is scored on a five-point scale, with 1 for “not at all” and 5 for “completely”. The average number of items is used as the scale score, and the higher the score, the higher the level of self-esteem. The scale has been widely used in Chinese populations and proved to have good validity and reliability for Chinese adolescents [64]. In this study, the alpha coefficient of the scale was 0.91.

#### 2.2.3. School Connectedness Scale

School connectedness was measured by using the school connectedness subscale of the Chinese Positive Adolescent Development Scale by Chai et al. [65] The scale consists of six questions and is scored on a five-point scale, with 1 for “completely disagree” and 5 for “completely agree”, and the average of the items is used as the scale score, with higher scores indicating higher levels of school connectedness. In this study, the alpha coefficient of the scale was 0.84.

#### 2.2.4. Psychological Crisis Questionnaire

The psychological crisis questionnaire is divided into two dimensions: depression and non-suicidal self-harming, and the total score of the questionnaire is the sum of the scores of each dimension. The depression scale uses the depression subscale in the Self-Rating of Symptoms Inventory (SCL-90) compiled and revised by Derogatis (2010) [66], which consists of 13 items. The depression self-assessment was performed on a 5-point scale, with 1 for “none” and 5 for “severe”, and the higher the score, the worse the depression. The scale has been widely used in Chinese populations and proved to have good validity and reliability for Chinese adolescents [67]. The Adolescent Non-suicidal Self-harming Assessment Questionnaire developed by Wan Yuhui was used to assess non-suicidal self-harming [68]. The scale consists of 12 items and adopts Likert 5-level scoring criteria ranging from 0 (no) to 4 (always). The total score of the scale is calculated by adding the scores of each item. The higher the score, the more serious the self-harming behavior is. In this study, the alpha coefficient of depression was 0.84, and the alpha coefficient of non-suicidal self-harming was 0.93.

### 2.3. Data Analysis

First, descriptive statistics and correlation analyses of the variables were conducted. Then, Mplus7.0 was used to test the mediating effects of self-esteem between controlling parenting styles, rejecting parenting styles, and psychological crisis, as well as the moderating effect of school connectedness on the mediating role of self-esteem. Since the estimates of the mediating effect usually do not follow the normal distribution, bootstrapping was used for testing the significance of the mediated effects and to produce percentile confidence intervals. If the interval does not contain 0, the mediation effect is significant. If the interval contains 0, then the mediation effect is not significant. Variables other than demographic variables were standardized before mediating effects and moderated mediating effects were analyzed. Several studies conducted in Hong Kong have shown that non-suicidal self-harming behaviors are more prevalent among females [69,70,71]. Therefore, the three variables of gender, rejecting parenting style, and controlling parenting style were included in the model in turn.

## 3. Results

### 3.1. Descriptive Statistics and Correlation

Descriptive statistics and bivariate correlations for all variables are reported in Table 1. The results of the correlation analysis indicated that the correlation coefficients of variables were statistically significant. Specifically, the rejecting parenting style and the controlling parenting style were negatively correlated with self-esteem and school connectedness, and positively correlated with psychological crisis. Self-esteem was positively correlated with school connectedness and negatively correlated with psychological crisis. School connectedness was negatively correlated with the rejecting parenting style, the controlling parenting style, and psychological crisis.

### 3.2. Regression Analysis

Stratified regression analyses were used to explore the predictive effects of parenting styles on psychological crisis among adolescents by incorporating demographic variables, rejecting parenting styles, and controlling parenting styles sequentially in the model. The results (see Table 2) showed that the inclusion of the rejecting parenting style in the second level (β = 0.46, *t* = 22.33, *p* < 0.001) positively predicted psychological crisis among adolescents. With the inclusion of the controlling parenting style in the third level, both the controlling parenting style (β = 0.15, *t* = 5.61, *p* < 0.001) and the rejecting parenting style (β = 0.37, *t* = 14.40, *p* < 0.001) were able to positively predict psychological crisis in secondary school students, but the rejecting parenting style was more predictive of psychological crisis than the controlling parenting style.

### 3.3. The Mediating Model

The mediating effect was tested by using the Bootstrap method in Mplus7.0. Bootstrap sampling was set at 5000 times, and the analyses were conducted sequentially with the rejecting parenting style and the controlling parenting style as the independent variables, self-esteem as the mediator variable, and psychological crisis as the outcome variable. The results showed that the mediating effect of the rejecting parenting style (β = 0.106, SE = 0.016, 95% CI = [0.077, 0.141]) affecting psychological crisis through self-esteem was significant. Similarly, the mediating effect of the controlling parenting style (β = 0.095, SE = 0.01, 95% CI = [0.066, 0.131]) affecting psychological crisis through self-esteem was also significant.

To further explore the co-effect of the two variables on self-esteem and psychological crisis, path analyses were conducted by placing rejecting parenting styles and controlling parenting styles into the same model. The results of the mediation model are presented in Figure 1. The results suggest that self-esteem still mediates the joint effect of the rejecting parenting style (β = 0.087, SE = 0.015, 95% CI = [0.060, 0.121]) and the controlling parenting style (β = 0.026, SE = 0.013, 95% CI = [0.004, 0.054]) on psychological crisis. The mediating effect of the rejecting parenting style and the controlling parenting style on psychological crisis through self-esteem was significant.

### 3.4. The Moderated Mediating Model

The moderating role of school connectedness on the mediating effect of self-esteem was examined according to the theoretical hypotheses of the present study. The results are shown in Figure 2. School connectedness (β = 0.346, *p* < 0.001) had a significant predictive effect on self-esteem, and the interaction term between the rejecting parenting style and school connectedness had a significant predictive effect on self-esteem (β = −0.072, *p* < 0.05). However, the interaction term between the controlling parenting style and school connectedness had no significant predictive effect on self-esteem (β = 0.017, *p* > 0.05). These results suggest that school bonding buffered the relationship between rejecting parenting styles and self-esteem, but did not buffer the direct relationship between controlling parenting styles and self-esteem. On the other hand, school connectedness (β = −0.318, *p* < 0.001) and self-esteem (β = −0.152, *p* < 0.001) were significant predictors of psychological crisis, and the interaction terms between self-esteem and school connectedness were also significant predictors of psychological crisis (β = 0.165, *p* < 0.001).

To more clearly explain the essence of the interaction effect, school connectedness was divided into high and low groups according to the standard deviation above and below the mean, and the moderating effect plots are shown in Figure 3 and Figure 4. Simple slope tests showed that, for adolescents with high school connectedness, the extent of the effect of rejecting parenting styles on self-esteem showed a significant decrease (βsimple = −0.256, *t* = 0.052, *p* < 0.001), whereas, for adolescents with low school connectedness, the negative effect of rejecting parenting styles on self-esteem was still significant but slowed down (βsimple = −0.111, *t* = 0.043, *p* < 0.001). Meanwhile, for the adolescents with low school connectedness, the influence of self-esteem on psychological crisis was significant and showed a significant downward trend (βsimple = −0.419, *t* = 0.076, *p* < 0.001). For adolescents with high school connectedness, self-esteem had no significant effect on psychological crisis (βsimple = −0.090, *t* = 0.053, *p* > 0.05).

## 4. Discussion

Adolescence is a critical period of individual development, but the gap between physical maturity and psychological development causes adolescents in this period to be at a higher risk for psychological crisis related to depression and self-harming [72,73]. Based on ecosystem theory, the susceptibility model, and the social support buffer model, this study systematically examines the joint effects of the rejecting parenting style and the controlling parenting style on adolescent psychological crisis, as well as the effects of self-esteem and school connectedness. Under the perspective of ecosystem theory, a moderated mediation model was constructed with self-esteem as the mediating variable and school connectedness as the moderating variable, which not only revealed how the two negative parenting styles affect the formation of adolescent psychological crisis (the mediating role of self-esteem) but also responded to the question of under which conditions parenting styles are more significant in affecting adolescent psychological crisis (the moderating role of school connectedness).

This study found that both rejecting and controlling parenting styles directly predicted psychological crisis in adolescents, which is consistent with the research hypotheses and the results of previous studies [27,28]. That is, if parents reject or overprotect their children for a long time, it can create negative psychological perceptions [74] which can lead to despair, distress, and then psychological crisis. Based on previous studies, this study found that the rejecting parenting style has a greater impact on psychological crisis than the controlling parenting style. In general, cultural values that encourage individual autonomy are often seen as part of individualistic cultures, whereas collectivistic cultures emphasize individual respect and obedience to authority [75]. Collectivist cultures place more emphasis on bonding among family members than Western individualistic cultures. As a result, the impact of controlling (overprotective) parenting styles on children’s psychological crisis is diminished. Therefore, for Chinese adolescents, research on the effects of parenting style on psychological crisis must be integrated into the social culture. In this study, psychological crisis did not show a gender difference. This may be due to the fact that the self-harming questionnaire used in this study contained too many ways of self-harming to reflect differences overall.

### 4.1. Mediating Effects of Self-Esteem between Negative Parenting Styles and Psychological Crisis

The present study verified the partial mediating role of adolescents’ self-esteem between rejecting parenting styles and psychological crisis, which is consistent with the findings of previous research [76]. The basic needs theory [77], derived from the self-determination theory, states that the rejecting parenting style makes it difficult for adolescents to establish secure attachments and positive interactions with their parents. With unmet needs of belonging, they feel unaccepted and worthless, and in turn, they develop negative perceptions of themselves [78]. As a result, the rejecting parenting style often makes adolescents feel inferior and gives them lower self-esteem when they interact with their peers, leading to psychological crisis. In addition, this research has also found that self-esteem mediates the relationship between controlling parenting styles and adolescent psychological crisis, which is consistent with previous research results [79]. According to self-determination theory [26], adolescents are at the stage of “separation-individuation”, and developing the need for autonomy and a sense of independence is an important task in this stage [80]. Parental psychological control limits adolescents’ freedom, threatens their need for autonomy, and reduces their sense of self-efficacy [81,82]. As a result, parents with high levels of control tend to make their children feel more frustrated and have lower self-esteem, leading to psychological crisis.

Further analysis of the joint effect of the two independent variables on self-esteem and psychological crisis in one model revealed that the mediating effect of the rejecting parenting style on psychological crisis through self-esteem is more significant, and in combination with the results of the hierarchical regression analyses, the rejecting parenting style was a stronger predictor of psychological crisis. This is reflected in the fact that rejecting parenting styles are more likely to affect adolescents’ self-esteem levels than controlling parenting styles [83], which in turn are more likely to exacerbate psychological crisis. The results of this study also support the susceptibility model of psychological crisis, which states that the susceptibility factor (self-esteem) can mediate the relationship between the stress factor (negative parenting style) and psychological crisis.

### 4.2. The Moderating Role of School Connectedness on the Mediating Effects of Self-Esteem

Firstly, when analyzing how school connectedness affects the effects of both parenting styles on self-esteem in the same model, it was found that the “relationship between rejecting parenting and self-esteem” was moderated by school connectedness, but that the relationship between controlling parenting and self-esteem was not affected by school connectedness. According to social support theory [35], school connectedness serves as a protective factor in the environment to mitigate the negative effects of adversity on adolescents. School connectedness has been found to have a buffering effect on adolescents exposed to adverse family risk factors, with positive school connectedness mitigating environmental threat factors and reducing the negative impact of family dysfunction on adolescents [84].

On the one hand, the self-esteem of high-school-connectedness individuals is less impacted by rejecting parenting styles than that of low-school-connectedness adolescents. That is, individuals with high school connectedness have access to higher levels of social support and attention, which can compensate for adolescents’ unmet needs from their families [85], facilitating individuals’ sense of self-efficacy and worth and maintaining self-esteem [52]. In contrast, due to the lack of necessary psychosocial resources, adolescents with low school connectedness feel that they are not accepted by their families and other groups, which is detrimental to the individual’s ability to build a favorable image, and the level of self-efficacy and self-esteem cannot be effectively enhanced. On the other hand, it was found that the relationship between controlling parenting style and self-esteem was not affected by school connectedness. This is because the controlling parenting style affects a child’s sense of autonomy and competence [86], and the school connection does not diminish the impact of this negative parenting style on a child’s sense of competence and self-esteem. In addition, Chinese parents place a strong emphasis on “training” in parenting, emphasizing the need to monitor and control their children [87]. In the view of Chinese parents, monitoring and controlling their children is not only necessary but also their responsibility. Therefore, if adolescents interpret parental control as care or control, their emotional needs will be met [88], and they will not have to compensate for their unmet parental needs through school connection.

In addition, this study found that school connectedness moderates the pathway of self-esteem influencing psychological crisis regardless of rejecting parenting style or controlling parenting style. Compared with the adolescents with high school connectedness, the self-esteem of the adolescents with low school connectedness has a more obvious negative prediction effect on psychological crisis. The reason for this may be that individuals with low school connectedness perceive themselves as socially inadequate, leading to lower self-evaluations and self-efficacy, and thus feel more depressed [89]. In contrast, for individuals with high school connectedness, a psychological crisis rises slowly with the decrease in self-esteem. This suggests that school connectedness can alleviate the direct impact of self-esteem on psychological crisis; even if the level of self-esteem is not high, it will not have a direct impact on individual psychological crisis. The results support the social support buffer model.

### 4.3. Implications and Future Prospects

This study found that negative parenting styles can influence the occurrence of adolescent psychological crisis. Therefore, changing parents’ negative parenting styles is an important way to mitigate adolescent psychological crisis. In recent years, intervention programs such as the attachment-based family therapy model, positive thinking parenting, and parental efficacy system training courses have gradually received widespread attention from educators and practitioners and have achieved good results [90,91,92]. Therefore, schools can encourage more parents to participate in such intervention programs and try to improve their negative parenting styles to reduce the occurrence of students’ psychological crisis. At the same time, it has been found that school connectedness can mitigate the effects of negative parenting styles on psychological crisis. Therefore, schools can conduct interpersonal-relationship-themed programs and empathy training to enhance interpersonal connections among peers and guide adolescents to establish peer support systems to mitigate psychological crisis. However, there are some shortcomings in this study: the cross-sectional study method cannot reflect the causal relationship to a certain extent, and future studies may consider using a larger sample size to observe the dynamic relationship of the above variables over time by using tracking data at multiple time points or the Hierarchical Linear Model (HLM) method. In addition, the data collected in this study came from adolescents’ self reports, but depression and self-harming behaviors have stigmatizing characteristics that are not accepted by society, so self reports may have problems of subjectivity and reliability. Future research could collect data from multiple sources such as parents, teachers, and peers.

## 5. Conclusions

(1)Compared with controlling parenting styles, the rejecting parenting style has a more serious negative impact on adolescents’ psychological crisis, which is partly achieved through adolescents’ perceived lower self-esteem.(2)School connectedness plays a moderating role in both “rejecting parenting style →self-esteem” and “self-esteem→psychological crisis”, whereby school connectedness can effectively alleviate the impact of the rejecting parenting style on self-esteem, and can enhance adolescents’ self-esteem to reduce psychological crisis.

## Figures and Tables

**Figure 1 behavsci-13-00929-f001:**
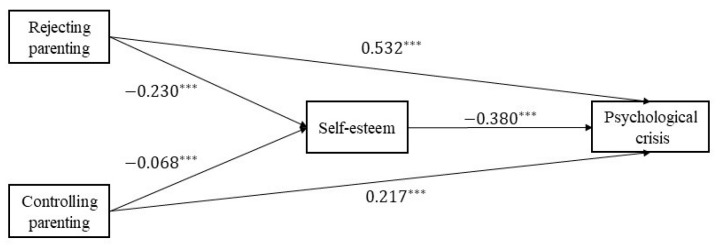
The mediating effect of self-esteem. Note. *** *p* < 0.001.

**Figure 2 behavsci-13-00929-f002:**
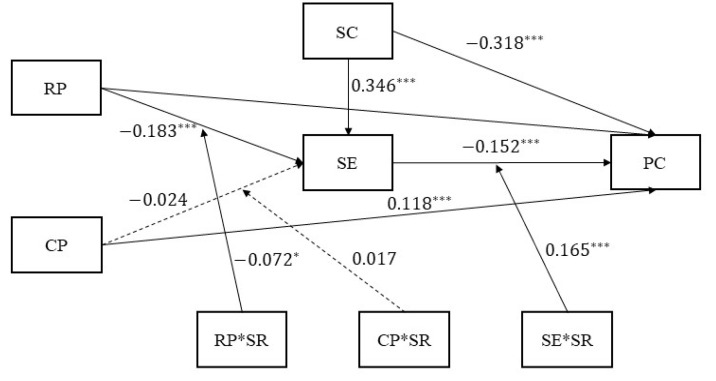
The moderating effect of school connectedness on the self-esteem mediating effect. RP = parenting style; CP = controlling style; SE = self-esteem; SR = school connectedness; PC = psychological crisis. Note. * *p* < 0.05; *** *p* < 0.001.

**Figure 3 behavsci-13-00929-f003:**
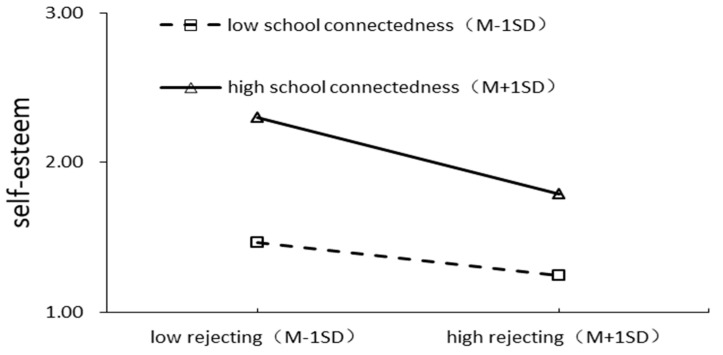
The moderating effect of school connectedness on the relationship between rejecting parenting style and self-esteem.

**Figure 4 behavsci-13-00929-f004:**
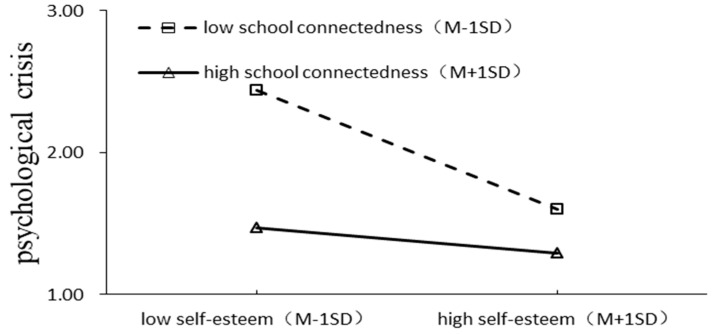
The moderating effect of school connectedness on the relationship between self-esteem and psychological crisis.

**Table 1 behavsci-13-00929-t001:** Descriptive statistics and bivariate correlations of the main variables.

	M	SD	1	2	3	4	5
Rejecting parenting	1.47	0.59	1				
Controlling parenting	1.83	0.74	0.61 **	1			
Self-esteem	3.03	0.59	−0.27 **	−0.21 **	1		
School connectedness	3.97	0.77	−0.28 **	−0.24 **	0.39 **	1	
Psychological crisis	0.00	1.67	0.46 **	0.37 **	−0.34 **	−0.36 **	1

Notes. ** *p* < 0.01.

**Table 2 behavsci-13-00929-t002:** Regression analysis of influencing factors on adolescent psychological crisis.

	First Level		Second Level		Third Level	
	*β*	*t*	*β*	*t*	*β*	*t*
Gender	0.03	1.34	0.03	1.51	0.03	1.54
Rejecting parenting			0.46	22.33 **	0.37	14.43
Controlling parenting					0.15	5.72
R^2^	0.001		0.14		0.23	

Notes. ** *p* < 0.01.

## Data Availability

Data are contained within the article and can be made available upon request.

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
