# Peer review of "Negative Parenting Styles and Psychological Crisis in Adolescents: Testing a Moderated Mediating Model of School Connectedness and Self-Esteem"

_behavsci, 2023, doi:10.3390/bs13110929_

Round 1

Reviewer 1 Report

Comments and Suggestions for Authors

Author Response

Thank you very much for your suggestion, please see the attachment for details.

Reviewer 2 Report

Comments and Suggestions for Authors

Authors report a study they conducted to test the moderating and mediating effects of school correctness and self-esteem on the psychological crisis in adolescents resulting from negative parenting styles. Authors found that both rejecting and controlling parenting style can lead to psychological crisis in adolescents, with rejecting styles producing a more serious negative impact. School connectedness moderates both the effects of rejecting parenting styles on self-esteem and the effects of self-esteem on the psychological crisis. Self-esteem partially mediates the relationship between rejecting parenting style, controlling parenting style, and psychological crisis.

Authors have provided strong justification for their hypotheses and study design through a well written introduction, which is then followed by thorough and appropriate quantitative analyses. Authors have cited appropriate and recent references through the manuscript.

I do not have any major revisions to suggest. However, I have few minor concenrs and sugegstions which are listed below.

1. Title could be rephrased from “Negative….: Testing a Moderated Mediating Model of Self-esteem and School Connectedness” to “Negative….: Testing a Moderated Mediating Model of School Connectedness and Self-esteem” as school corrected has been shown to play moderating effect and self-esteem play mediating effect.

2. Methods sections, 2.1 Participants and Procedure lacks information about the timing (which years) of the data collection, that is when the data was collected. For example, data collected during COVID pandemic or post-pandemic can have additional layers of complexity compared to if the data was collected before 2019. This information is critical for such a topic, please provide the timeline study in the methods section.

3. A significant limitation of studies that depend on self-reports from adolescents is that the accuracy of their comprehension regarding the true intent of the questions might be compromised due to their limited social and understanding ability. Please comment if authors took any precautions related to it or have suggestions for future studies.

4. Allowing parents to self-assess the extent of psychological control they exert over their children can yield intriguing and highly valuable insights for the formulation of parenting training programs. How do the authors perceive such potential avenues for future research?

 Overall, the current manuscript is presented in a well-structured manner. It has potential to contribute to the field of adolescent psychology and impacts of parenting styles whereby such a model-based study can help develop promising strategies to improve the parenting skills and ultimately, benefit the mental health of adolescents.

Author Response

Thank you very much for your suggestions, please see the attachment for details.

Reviewer 3 Report

Comments and Suggestions for Authors

Dear author congratulation for the conduction of this research analysing an important issue.

I've made some suggestion in order to improve the manuscript. Please see the attached file.

Author Response

(The authors gave the same response as above.)

Reviewer 4 Report

Comments and Suggestions for Authors

This study focuses on important aspects related to adolescents' mental health. The scope and the dimensions studied are interesting. However, some clarifications are needed. Critical points that require further work before considering the paper for publication are as follows:

- Better definition and measurement of psychological crisis: the definition of this construct seems to emerge from a combination of 2 variables. I am not sure about the fact that depression can be considered as the key indicator of adolescents' crisis. Anxiety is another core psychological problem adolescence. Self-harm can also be linked to other diagnosis, e.g. bipolar disorder. I wonder if it is better for the authors to consider the 2 dimensions separately, perhaps testing 2 models? One on self harm and one on depression? Also, the term psychological crisis is very generic, it can include trauma, transitional social-emotional problems. The authors are referring here to mental health issues. Better details are needed and stronger references to combine the 2 dimensions in the construct of psychological  crisis.

- References issues found - review referencing style throughout. There is no correspondence between in-text citation and references list.

- I suggest that the authors use the term self-harm instead of self-injuries.

- Also, please use 'participants' instead of 'subjects'

Comments on the Quality of English Language

Please proofread the article.

Author Response

(The authors gave the same response as above.)

Round 2

Reviewer 1 Report

Comments and Suggestions for Authors

Thanks for revising the manuscript. I appreciate that you have fully addressed my concerns in the reviewer's report.